# LLMServingSim: A Simulation Infrastructure for LLM Inference Serving Systems

Jaehong Cho, Minsu Kim, Hyunmin Choi, Jongse Park

KAIST. {jhcho, mskim, hmchoi, jspark}@casys.kaist.ac.kr

*Abstract*—Recently, there has been a large research effort in building efficient large language model (LLM) inference serving systems, including advancements in both hardware and software. Nevertheless, there is a lack of simulation infrastructure capable of accurately modeling hardware-software system behaviors without extensively extending simulation time. This paper aims to solve the limitations of existing system simulators and develop an effective simulation tool, called LLMServingSim, to support future research in LLM inference serving systems. In designing LLMServingSim, we focus on two algorithmic properties: (1) the dynamic variation in workload characteristics of LLM inference serving due to its autoregressive nature, and (2) the need for detailed memory modeling due to the large key-value (KV) cache generated during runtime inference serving. This paper describes the key challenges contributing to bridge the "real2sim" gap and presents our initial strategies to address them. It also discusses the unresolved problems that persist.

## I. INTRODUCTION

Currently, there is a significant surge in efforts to exploit large language model (LLM) as a crucial component in real-world applications [16], [39]. Given the prohibitively high costs associated with building on-premise infrastructure for LLM inference, the common practice is to offload LLM inference to multi-tenant "inference serving" systems in the cloud, exemplified by OpenAI's ChatGPT service [19]. The massive compute and memory requirements (both bandwidth and capacity) are forcing these systems to be equipped with many AI accelerators (or NPUs) that typically come with high-bandwidth memory stacks (e.g., NVIDIA H100 [17]).

There has been a large body of research works that aim to develop efficient hardware and software for LLM inference serving systems. Some works target to develop customized hardware techniques for accelerating LLM inference serving [9], [21], while others focus on developing optimized system software on GPU-based scale-out systems [5], [12], [13], [18], [23]. Recently, a few pioneering works propose to take into consideration both hardware and software together for designing holistic end-to-end accelerated systems [8], [20]. However, there is currently a lack of simulation infrastructure that allows researchers to explore their hardware-software proposals in a scale-out setting. This limitation not only makes it difficult for computer architecture researchers to explore scalable accelerator solutions, but also leads computer system researchers to concentrate on GPU-based system software in the era of specialized hardware.

This paper sets out to address this limitation and develop a LLM inference serving system simulator, called LLM-ServingSim, that jointly simulates the behaviors of LLM-customized accelerators and LLM inference serving system software. LLMServingSim is built on top of an existing AI system simulator, ASTRA-sim [36], which jointly models both hardware and software for AI workloads. However, there are primarily two algorithmic differences, making the design principles of LLMServingSim and ASTRA-sim largely different, as described below.

- **Autoregressive nature of LLM generation.** ASTRA-sim focuses on distributed training, which entails millions of "identical" iterations of computing that simplify the simulation. On the contrary, we target LLM inference serving that involves autoregressive token generations, producing dynamically changing behaviors across different iterations, requiring independent simulation runs for them.
- **Large KV cache generated at runtime.** ASTRA-sim lacks detailed memory modeling since the memory requirements are statically determined at the compile time. However, LLM inference serving produces large KV cache during inference serving at runtime, necessitating the memory modeling for accurate simulation.

To this end, we design LLMServingSim in such a way that it prudently compromises simulation accuracy for achieving the feasible simulation time, effectively bridging the so-called "real2sim" gap, and in turn, facilitating the future research in LLM inference serving systems. To accomplish these objectives, we exploit the following three major techniques.

- **Iteration-level hardware-system simulation.** As each iteration takes different input prompts, LLMServingSim simulates the iterations one by one temporally and aggregates the entirety of resulting statistics at the end. For each iteration, LLMServingSim first performs prompt scheduling that determines tasks for accelerators, then analyzes the accelerator behaviors using hardware simulator, and finally sweeps through the stages in the system pipeline to simulate overall system behaviors. For hardware simulator, we employ GeneSys [7], an open-source end-to-end NPU simulator that comes with a full software stack. The aforementioned three steps are repeated over the iterations progressively.
- **KV cache-aware detailed memory modeling.** The nature of LLM token generation and its reliance on KV cache necessitate LLMServingSim to have a detailed memory modeling. LLMServingSim employs a state-of-the-art memory management scheme for LLM inference serving system, demand paging [12], and maintains the application states and system-level statistics, such as generated tokens and

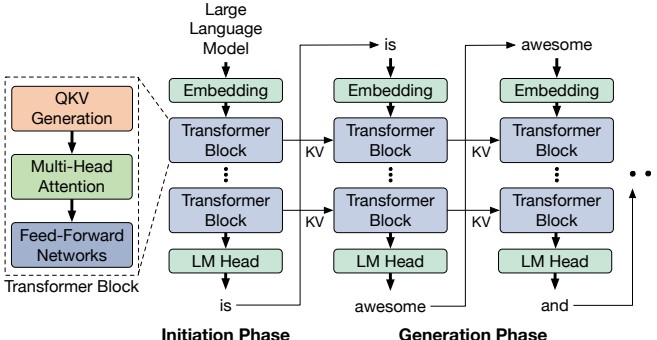

Fig. 1. Architecture of large language model.

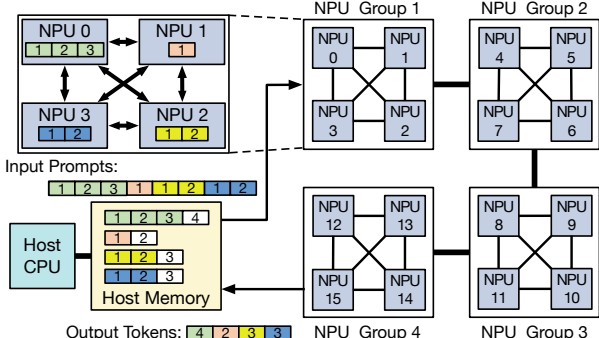

Fig. 2. Topology of LLMServingSim serving system architecture configured with hybrid parallelism, consisting of 4 NPU groups and 16 NPU nodes.

KV cache size.

- **Compiler and simulator optimization through computation reuse.** We notice that the hardware simulator, along with the aforementioned two techniques, experiences a substantial bottleneck at the compilation and hardware simulation phases. LLMServingSim addresses this bottleneck by optimizing implementations exploiting the redundancy of common LLM architecture and employing computation reuse techniques.

Our experiments demonstrate that the simulation results produced by LLMServingSim show a similar trend as in the real LLM inference serving system, while offering a feasible simulation time that scales up to a few hundreds of seconds. These promising results suggest that LLMServingSim has a significant potential to be an effective system simulation tool for LLM serving system research, in hardware, software, or both.

## II. BACKGROUND

### A. Characteristics of LLM Model Architecture

Most large language models (LLMs), whose architecture is depicted in Figure 1, employ decoder-based transformer structure [35]. This architecture is constructed with its fundamental building units: embedding layer, transformer blocks, and language modelling (LM) head. Each transformer block consists of three main components: Query, Key, Value (QKV) generation, multi-head attention, and feed forward networks.

Decoder-based transformer model operates in two distinct phases during their inference: the initiation phase and the generation phase. The initiation phase begins with receiving the prompt as input and generates QKV for all input tokens. Generated QKV passes through subsequent multi-head attention layer and feed-forward networks. Once the initiation phase is completed, the model outputs one token and transitions to the generation phase with the generated token as new input of the model. The generation phase has autoregressive characteristic where each output token is passed to the next iteration and the generation continues sequentially. In this phase, QKV for newly generated tokens needs to be computed while utilizing cached key-value of previous tokens, known as KV cache.

### B. Batching and Memory Management for LLM Serving

To minimize latency and maximize hardware utilization, LLM inference serving system often employs the concept of batching, which involves grouping multiple requests into a single group. However, it presents a challenge, particularly with the multi-head attention layer, which makes batching difficult. Additionally, it faces the drawback of needing to complete all requests before proceeding to the next batch, which can lead to inefficiencies. Orca [37] tackles this challenge through selective batching and iteration-level scheduling. Selective batching allows batching in specific layers, such as QKV generation and feed-forward networks, while in multi-head attention layers, it allow a batch to be divided and allocated to multiple workers individually. Iteration-level scheduling involves rescheduling the batch at each iteration, removing completed requests and adding new ones. This technique enhances hardware utilization and reduces latency by dynamically updating the batch to include only active requests, thereby streamlining the process.

Another challenge in scale-out inference serving system is to effectively handle KV cache. Conventional LLM inference serving allocates KV cache based on the maximum possible sequence length, and this results in underutilized memory spaces and limited batch sizes. vLLM [12] introduces a paging scheme for memory management that operates similarly to the virtual memory of operating systems. Managing memory on a page-by-page basis, vLLM effectively reduces memory fragmentation, enabling larger batch size and higher throughput.

## III. LLMSERVINGSIM

### A. Overview of LLMServingSim

LLMServingSim jointly simulates LLM inference serving system software and hardware optimized to accelerate LLM inference workload. In this paper, for simplicity of explanation, we give an example where LLMServingSim simulates a distributed system consisting of one host node and multiple NPU nodes. We assume that the host node consisting of CPU and DRAM runs the LLM inference serving system software and orchestrates the NPU nodes, while the NPU nodes have relatively small device memory and execute the LLM inference operations. As in the widely adopted LLM

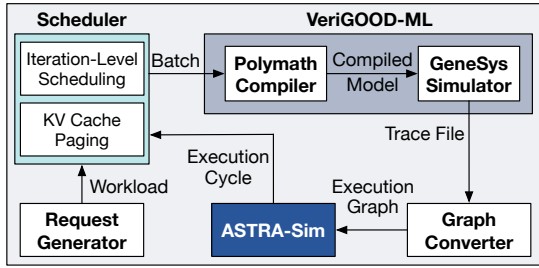

Fig. 3. Workflow of LLMServingSim.

serving system, LLMServingSim supports several parallelism strategies. Figure 2 illustrates the LLMServingSim system topology configured to utilize hybrid parallelism with 4 NPU groups and 16 NPU nodes.

Figure 3 depicts the LLMServingSim workflow, which is designed to perform iteration-level hardware-system simulation. Furthermore, LLMServingSim employs the memory management scheme introduced in the state-of-the-art LLM serving system, vLLM. LLMServingSim consists of the following components: **(1) Scheduler** receives user requests and organizes them into feasible batches based on the scheduling and KV cache management strategy. It makes next scheduling decision based on the results of ASTRA-sim. **(2) PolyMath compiler** [11] compiles the models according to the batch configuration created by Scheduler. **(3) GeneSys simulator** [7] performs hardware simulation for a single NPU and produces output trace files from the compiled models. **(4) Chakra graph converter** uses the trace files to create execution graphs for each NPU in the system according to the configured parallelism strategy. **(5) ASTRA-sim** [36] takes Chakra graphs [33] as inputs, performs system simulation, and returns results back to the scheduler. The following sections introduce the techniques one by one.

### B. Iteration-level Scheduling for Hardware-System Simulation

LLM processes input prompts autoregressively by generating one token at a time during inference. To efficiently process these iterations, Orca [37] proposes iteration-level scheduling. Inspired by Orca, we employ this approach in LLMServingSim by designing the simulation workflow as repeated alternations of prompt batch scheduling, hardware simulation, and system simulation at the iteration level.

LLMServingSim scheduler first receives requests and compares their arrival times to the scheduler's timer to select *batchable* requests. In response to the dynamic changes in requests, the scheduler leverages the Polymath compiler and GeneSys to simulate the behavior of the hardware accelerators. They compile the model and simulate the hardware with specified input configurations. After hardware simulation, Chakra graph converter converts the simulation result to a graph that maps the hardware to the system. This graph is then fed into ASTRA-sim to simulate and analyze the system behavior comprehensively. System simulation results are fed back to the scheduler, and the scheduler's timer, which is used to assemble

a new batch for the next iteration, is updated accordingly. This cyclical interaction enables LLMServingSim to progress through iterations efficiently and bridge the "real2sim" gap.

### C. Supporting for LLM Parallelism Strategies

In the context of LLM inference, parallelism that distributes the model weights and layers of substantial size is crucial for enhancing the inference performance. There are three major types of model parallelism: tensor parallelism, pipeline parallelism, and hybrid parallelism [32].

LLMServingSim can be configured to utilize a specific parallelism and the number of NPU groups to determine the topology of the system. When Chakra graph converter receives the output trace from GeneSys simulator, it identifies configured parallelism strategy and constructs execution graph accordingly for each NPU. In the case of tensor parallelism, it distributes tensors across the entire NPU and inserts ALL-REDUCE operators to the graph for intermediate synchronization. In the case of pipeline parallelism, it allocates decoder blocks to NPUs in sequence, allowing chained computation across NPUs. For hybrid parallelism, it combines both parallelism strategies by distributing tensors and layers within and across NPU groups, respectively. Consequently, generated Chakra graph has both aspect of inter-group pipeline parallelism and intra-group tensor parallelism.

To employ selective batching, where attention layers are processed in parallel across different workers, GeneSys simulator and Chakra graph converter work in conjunction. GeneSys simulator assigns unique identifiers to the attention layers within each batch and records them in the output trace. Chakra graph converter then assigns these attention layers to different NPUs based on their identifiers. As illustrated in Figure 2, within an NPU group, each NPU independently processes distinct inputs with different sequence length, efficiently parallelizing batch processing.

### D. KV Cache-Aware Detailed Memory Modeling

While ASTRA-sim has a simple memory model in its implementation, it lacks some memory constraints such as capacity and memory fragmentation. However, LLM inference is sensitive to memory capacity due to their significant memory usage of model weights and KV cache. LLMServingSim uses detailed memory modeling scheme with several memory constraints to reduce the gap with actual systems. Memory model of LLMServingSim includes management of KV cache and generated tokens by incorporating demand paging technique from vLLM [12].

The management for KV cache and generated tokens in LLMServingSim scheduler is intertwined with iteration-level scheduling, which conducts batch reconstruction each iteration, by checking generated tokens and KV cache size of each batch. First, scheduler assesses the length of incoming requests to determine the required number of KV cache pages and allocates them to the device memory accordingly to form a single batch. After an iteration completes, the scheduler reassesses the requests. If increased sequence length due to

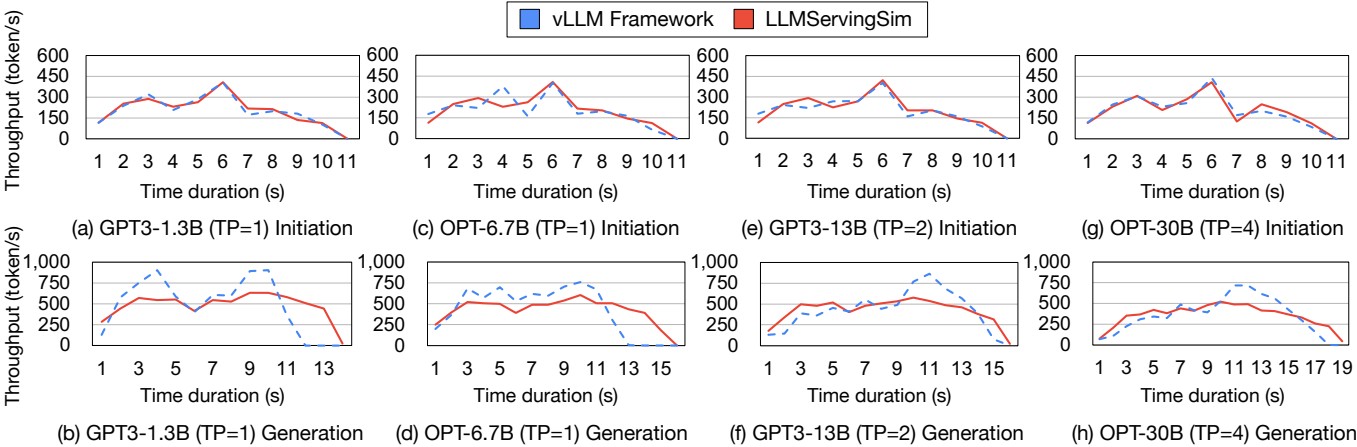

Fig. 4. Comparison of throughput over time between GPU-based LLM serving system and LLMServingSim using request pattern following a Poisson distribution.

generated tokens requires additional page or incoming requests need be added to the batch, new page is allocated on demand. If there is insufficient memory capacity for new pages, the entire page for KV cache and the sequence of the lastly added requests are evicted to host memory.

The graph converter inserts operators into the execution graph for page eviction and reloading based on the decision of the scheduler. Whenever page eviction or reloading occurs, it inserts memory store or load operator nodes embedded with the time taken to transfer the pages between device memory and host memory into the graph. This interaction between the scheduler and Chakra graph converter enables LLMServingSim to effectively utilize page-based memory modelling.

### E. Simulation Time Optimization through Computation Reuse

Given the large size of LLM, which typically results in lengthy compile and hardware simulation time, we introduce optimization techniques with computation and redundancy reuse. First, we achieve significant time savings by exploiting the redundancy of common LLM architecture. As described in Figure 1, decoder-based LLM architecture consists of an embedding layer followed by repeated transformer blocks. LLMServingSim compiles just one transformer block and replicates it, significantly reducing the overall compile time required.

Another optimization to reduce simulation time involves separating attention layers from non-attention layers. The initiation phase and the generation phase differ only in attention layers, depending on the presence or absence of KV cache. Therefore, LLMServingSim compiles and simulates the time-consuming non-attention layers just once, and subsequently, it simply swaps out the less time-intensive attention layers, thereby cutting down on the total processing time.

Given the dynamic nature of input and output lengths in LLM inference, models typically need to be continuously compiled and simulated. LLMServingSim adopts a strategy of reusing previously simulated results, and for effective caching,

it manages the non-attention layer and attention layers differently. Non-attention layers take longer than other layers to be processed but can be reused frequently. However, attention layers require more frequent compilation and simulation but take less time. We conduct an evaluation to evaluate the impact of this caching strategy and demonstrate that our optimization technique is effective in reducing the overall simulation time.

## IV. EVALUATION

### A. Methodology

Throughout our evaluation, we use a GPU system equipped with 4 NVIDIA RTX 3090 GPUs with 24GB VRAM and Intel Xeon Gold 6326 CPU as the actual inference serving system baseline. We use vLLM [12] framework as LLM inference serving system software. For running LLMServingSim, we use a CPU system equipped with an Intel Xeon Gold 6226R CPU with 96GB DRAM. We configure the hardware architecture of the NPU in LLMServingSim as a 128x128 systolic array with a clock speed of 1GHz. The device memory bandwidth is set to match that of GPU at 936GB/s, and the NPU-NPU link bandwidth is set to be equivalent to PCIe 4.0 ×16 bandwidth at 64GB/s.

### B. Simulator Validation

We evaluate how accurately LLMServingSim simulates the actual LLM inference serving system. Figure 4 shows the fluctuation in throughput of request serving over time in a dynamic request pattern for the GPT-3 [4] and OPT [38] models, with sizes varying from 1.3B to 30B. For the workload, we sample requests from ShareGPT [30] and synthesize request arrival pattern using Poisson distribution. We set the tensor parallelism degree from 1 to 4 depending on the model size and set the memory capacity to 24GB to match that of GPU.

In the throughput trend of initiation phase, as shown in the upper row of Figure 4, we observe a high degree of similarity in the initiation throughput trends between LLMServingSim and GPU-based system, resulting in a correlation coefficient of 0.93, which indicates existence of high correlation between

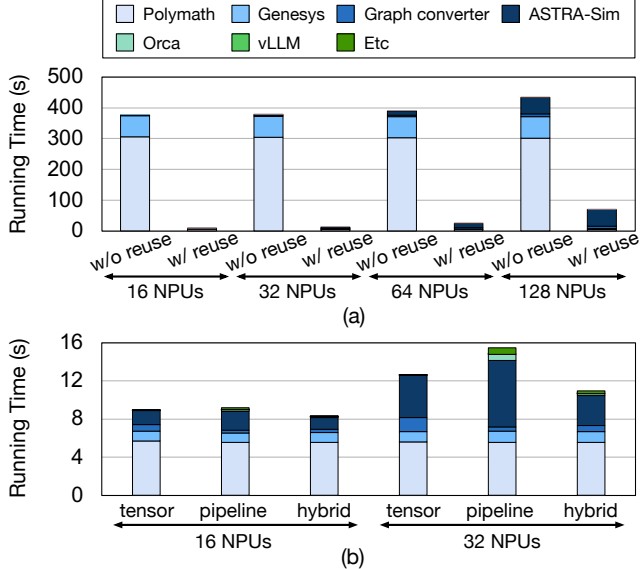

Fig. 5. Breakdown of LLMServingSim simulation time. (a) Comparison of simulation running time with and without computation reuse using varying numbers of NPUs. (b) Simulation running time of with computation reuse for three parallelism strategies.

them. Specifically, throughput of initiation phase is influenced not only by the scheduling decision to form a request batch but also by the system's ability to accommodate the incoming requests' KV cache in memory. Therefore, these trend results demonstrate that the iteration level prompt scheduling, and detailed memory modeling of LLMServingSim closely mirrors the behavior observed in GPU-based system.

The lower row of Figure 4 depicts the throughput trend in generation phase. We observe that LLMServingSim often follows the generation throughput trend of the GPU baseline, as indicated by high correlation coefficient of 0.79. However, unlike the trend observed in the initiation phase, there are some performance discrepancies, which can be attributed to several factors. First, it is challenging to configure NPU architecture to precisely match the performance of the GPU. Additionally, the degree of kernel operation optimization varies between GPU-based system and LLMServingSim. While GPU systems employ kernel optimization techniques such as FlashAttention [5], the absence of such kernel optimization in LLM-ServingSim leads to throughput differences, especially under request-intensive conditions. Despite these deviations, the overall throughput trend of LLMServingSim resembles that of the GPU, confirming that LLMServingSim can effectively reduce the "real2sim" gap.

### C. Simulation Time Breakdown

Figure 5 shows the entire simulation time and its breakdown to each component of LLMServingSim with various system configurations. In this measurement, we use GPT-3 175B model and input sequence length of 2048, and measure the simulation time to complete one iteration. Given the large

size of the model, we configure the NPU memory capacity to 40GB.

Figure 5(a) compares the simulator running times both with and without the computation reuse optimization with the varying number of NPUs from 16 to 128. As the number of NPUs increased, we observe a corresponding rise in the simulator's running time, increasing by 15.1%. The running time varies significantly depending on whether reuse optimization was utilized or not. Without reuse, running times range from 377.0 to 433.8 seconds, but with the optimization enabled, they range from 8.9 to 69.8 seconds, demonstrating a substantial speedup of 6.2× to 42.3×. Computation reuse eliminates the need to rerun the Polymath compiler and GeneSys simulator for each iteration. This highlights the significant performance benefits of computation reuse optimization applied to LLMServingSim.

Figure 5(b) also compares the simulator running time of each component using three parallelism strategies. We use 4 NPU groups for hybrid parallelism, regardless of the number of NPUs. ASTRA-sim's execution time is longest when using pipeline parallelism, as it requires simulating more cycles than other parallelisms due to the inherent long latency of pipeline parallelism. The running time of LLMServingSim scheduler, including vLLM and Orca, also increase with pipeline parallelism. Unlike the case of tensor and hybrid parallelism, where the scheduler waits until entire batch is processed, the scheduler makes a new decision at each pipeline stage, which adds additional overhead. While there is a variance in simulation time among LLMServingSim's three parallelism strategies, the difference is minimal. This allows for the simulation of various system configurations within a feasible time, facilitating hardware and software exploration for LLM inference serving system.

## V. Discussion

### A. Usability of LLMServingSim

**Pluggability to 3rd-party accelerators.** The LLMServingSim's architecture allows for high configurability within the GeneSys NPU framework, where elements of the NPU can be customized or swapped with ease. Furthermore, LLMServingSim supports the integration of various third-party accelerators. If these accelerators can generate traces that match the existing template, they can be added to the system. Beyond computational units, it is possible to extend memory features, for instance, by adding storage capabilities or incorporating advanced technologies such as Processing In Memory (PIM) or SmartSSDs. This flexibility makes LLMServingSim a highly versatile tool for simulation and development.

**Compatibility with existing machine learning frameworks.** LLMServingSim takes the ONNX [14] model format as an input, enabling compatibility with various machine learning frameworks. This compatibility allows users to seamlessly integrate widely-used open-source ONNX models. Additionally, models from frameworks such as PyTorch [22] and TensorFlow [1] can be converted to ONNX format for use

within LLMServingSim, facilitating a broad range of model experimentation and deployment scenarios.

### B. Limitations and Future Works

As the performance of LLM models improves, the size of model parameters also increases. Further, memory capacity requirements are rapidly increasing due to larger batches and longer sequence lengths. Prior works have proposed techniques to utilize not only device memory but also heterogeneous memory or storage, such as NVMe and flash memory [2], [25]. However, LLMServingSim currently supports only device memories and host memory in the memory hierarchy, and in the future, we plan to enhance support for inference of very large models by implementing various types of memory or storage.

## VI. RELATED WORKS

**Hardware simulators.** There are several well-known hardware simulators that precisely model hardware behavior for ML workloads. Several simulators aim to simulate ML operations cycle accurately by developing single-core systolic array-based accelerator simulator [28], [29]. Additionally, other studies endeavor to model interactions between cores in a multi-core NPU architecture [6], [10], [24]. LLMServingSim specifically targets the accurate modeling of LLM inference workloads, which exhibit distinct algorithmic and memory patterns compared to conventional ML workloads. Extending beyond existing studies that concentrate on simulating behavior within a single NPU chip, LLMServingSim devises a mapping scheme for LLM workloads in multi-NPU environment.

**System simulators.** In the realm of distributed system simulators, tools have been developed to cater to a range of needs from general-purpose workload simulators [15], [27], [34] to those specifically designed for neural networks [26], [31], [36]. Recently, more specialized simulators tailored for LLM training have emerged [3]. While existing solutions struggle to overcome challenges in simulating LLM inference serving systems, LLMServingSim successfully tackles them by exploiting techniques including iteration-level scheduling, KV cache paging, and the interaction between hardware and system simulators.

## VII. CONCLUSION

The absence of system simulator tailored for LLM inference, which possesses unique algorithmic traits, presents challenges for researchers in system or hardware architecture exploration. In this paper, we address these challenges by introducing LLMServingSim, a hardware-system simulator targeting LLM inference serving systems. To reduce "real2sim" gap, LLMServingSim incorporates a simulator design that considers the characteristics of LLM inference and the memory management scheme of the state-of-the-art LLM inference systems, while applying optimization to achieve feasible simulation times. In our evaluation using a real system with dynamic request patterns, we observe that the throughput trend of LLMServingSim closely corresponds to that observed in the real system.

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
