# OpenReview forum: "LLMServingSim: A Simulation Infrastructure for LLM Inference Serving Systems"
_iscaconf.org/ISCA/2024/Workshop/MLArchSys — MLArchSys 2024 OralPoster_

### Official Review · Reviewer_dmTq · 2024-05-24
**This paper introduces LLM-Sim, a simulation tool for accurately modeling hardware-software behaviors in LLM inference serving systems, addressing dynamic workload and memory usage challenges, but lacks clear quantitative evaluations of memory modeling impact and justification for NPU configurations.**

**Confidence:** 4
**Rating:** 7

**Detailed Feedback And Questions For Authors:**

This paper introduces LLM-Sim, a simulation tool designed to address the challenges of accurately modeling hardware-software system behaviors for large language model (LLM) inference serving systems. Current simulation infrastructures fail to accommodate the dynamic workload characteristics and detailed memory modeling required for LLM inference, leading to limitations in research and development. LLM-Sim overcomes these challenges by focusing on two key algorithmic properties: the dynamic variation in workload due to the autoregressive nature of LLM inference, and the detailed modeling of the large key-value (KV) cache generated during runtime. Built on the existing AI system simulator ASTRA-sim, LLM-Sim introduces iteration-level hardware-system simulation, KV cache-aware detailed memory modeling, and compiler and simulator optimization through computation reuse. These features enable LLM-Sim to provide accurate simulation results while maintaining feasible simulation times.

- It seems that one of the important contributions of LLM-Sim compared to ASTRA-Sim is the inclusion of memory modeling for accurate simulation, especially for LLM inference. Given this, if the evaluations of this paper could more clearly showcase the impact of such a memory model on making the overall simulation more accurate, it could add more value to this paper. Similar quantitative analysis could have been used in the motivation to emphasize the necessity of integrating an accurate memory model.

- The analysis related to Figure 4 includes very interesting observations. In particular, the authors point out the performance discrepancies and mention that one reason for this could be the configuration of NPU architecture to precisely match the performance of the GPU. Accordingly, the authors could have mentioned some justifications for why the current configurations are selected for NPUs and what specific trade-offs to consider for configuring NPUs.

**Top Reasons To Accept The Paper:**

LLM-Sim addresses the challenges of modeling dynamic workload characteristics and detailed memory usage in LLM inference serving systems, providing simulation results that can significantly aid future research.

**Top Reasons To Reject The Paper:**

The paper lacks clear quantitative evaluations demonstrating the impact of the detailed memory modeling on overall simulation accuracy, and does not adequately justify the selected configurations for NPUs, limiting the depth of its analysis.

---

### Official Review · Reviewer_WngB · 2024-05-28
**Good improvements on LLM inference system simulation but requires more discussion and comparison.**

**Confidence:** 3
**Rating:** 6

**Detailed Feedback And Questions For Authors:**

The paper is well-written and organized. The evaluation clearly shows good accuracy of the proposed system against real-world systems. The methodology is clearly described with figures.  The contribution proposed by this paper makes LLM inference systems more accurate and faster. I suggest the following additions to the paper.

1. What is the difference in the timing ratio for simulation and actual inference? Is the difference significant?

2. It would be better to include more motivation for simulation in the paper. What are the primary goals of this kind of simulation? Is the goal of simulation to reduce the time to evaluate the inference system, avoid using resources (i.e. GPUs) used to pre-evaluate the system?

3. Adding more discussion on what other levels of simulations are performed for LLMs and the trade-off among them would better show the importance of this work for the readers.

4. Can a direct comparison be made between LLM-Sim and ASTRA-Sim in similar settings as in Figure 4? That would give us some insights into the accuracy of LLM-Sim.

5. Are there any accuracy impacts due to the optimizations to improve simulation time like computation reuse?

**Top Reasons To Accept The Paper:**

LLM-Sim addresses a relevant problem of simulating contemporary LLM inference systems to explore hardware-software optimizations. The contributions include a detailed simulation of the memory system, modeling dynamic token generations of LLM models, iterative-level simulation and optimizations to reduce simulation time.

**Top Reasons To Reject The Paper:**

More discussion on trade-offs related to the simulation is required and comparison should be made against ASTRA-sim in similar settings.

---

### Official Review · Reviewer_dkUP · 2024-05-28
**The paper presents an LLM inference simulator for LLM serving systems, namely LLMSim.**

**Confidence:** 4
**Rating:** 6

**Detailed Feedback And Questions For Authors:**

The paper read well.

The LLM-Sim would be an interesting add to the LLM serving community for better understanding of system performance.

I would request the authors to add more serving scenarios.

I would also request to add with more models including those with GQA, MQA for KV cache sharing.

**Top Reasons To Accept The Paper:**

LLM-Sim jointly simulates LLM inference serving system software and hardware optimized to accelerate LLM inference
workload. LLM-Sim supports, a. Iteration-level Scheduling for Hardware-System Simulation, b. Support  LLM Parallelism Strategies, c. Models detailed memory with KV cache awareness. Thus the simulator handles major issues associated to LLM serving systems.

**Top Reasons To Reject The Paper:**

The paper does not compare with recent serving systems like Splitwise, Sarathi. It will be interesting to see LLM-Sim working on various serving secnarios.

---

### Official Review · Reviewer_w8F2 · 2024-05-28
**LLM-Sim, a hardware-system simulator targets LLM inference serving systems. LLM-Sim incorporates a simulator design that considers the characteristics of LLM inference and the memory management scheme of the state-of-the-art LLM inference systems, while applying optimization to achieve feasible simulation times. LLM-Sim exploits techniques including iteration-level scheduling, KV cache paging, and the interaction between hardware and system simulators.**

**Confidence:** 3
**Rating:** 7

**Detailed Feedback And Questions For Authors:**

In addition to above comments, I would like to ask if there is a future path for Jax based LLM models? Furthermore, would have liked to see any quantifiable comparison with existing state-of-the art inference simulation infrastructure.
There were some gaps in writing for example '“real2sim”' has been used at multiple places without any reference or explanation.

**Top Reasons To Accept The Paper:**

They have evaluated on multiple models and for different sized models which poses different memory/compute bounds showcasing efficacy of their simulation software. Furthermore their software takes ONNX which makes it usable for PyTorch and Tensorflow based models. They also optimizes simulation time resulting in 6.2× to 42.3× speedup.

**Top Reasons To Reject The Paper:**

- Their software will be limited to Jax based models
- They don't show comparison with any other LLM inference simulation software.

---

### Decision · Program_Chairs · 2024-05-30

**Decision:**

Accept (Oral/Poster)

**Comment:**

Congratulations! We are pleased to inform you that your paper has been accepted for presentation at MLArchSys 2024. We look forward to your participation at the workshop. Further details regarding the schedule and format will be provided soon. See you at the workshop!